# An Integrative Study on *Asphondylia* spp. (Diptera: Cecidomyiidae), Causing Flower Galls on Lamiaceae, with Description, Phenology, and Associated Fungi of Two New Species

**DOI:** 10.3390/insects12110958

**Published:** 2021-10-21

**Authors:** Umberto Bernardo, Francesco Nugnes, Simona Gargiulo, Rosario Nicoletti, Andrea Becchimanzi, Adriano Stinca, Gennaro Viggiani

**Affiliations:** 1Portici Unit, Institute for Sustainable Plant Protection, National Research Council (CNR), 80055 Portici, Italy; umberto.bernardo@ipsp.cnr.it (U.B.); francesco.nugnes@ipsp.cnr.it (F.N.); simona.gargiulo@ipsp.cnr.it (S.G.); 2Department of Agriculture, University of Naples ‘Federico II’, 80055 Portici, Italy; rosario.nicoletti@crea.gov.it (R.N.); andrea.becchimanzi@unina.it (A.B.); 3Council for Agricultural Research and Economics, Research Center for Olive, Fruit and Citrus Crops, 81100 Caserta, Italy; 4Department of Environmental Biological and Pharmaceutical Sciences and Technologies, University of Campania Luigi Vanvitelli, 81100 Caserta, Italy; adriano.stinca@unicampania.it

**Keywords:** *Botryosphaeria dothidea*, *Cladosporium*, *Clinopodium vulgare*, gall midge, *micromeriae*, *Micromeria graeca*, *Origanum vulgare*, *rivelloi*, *Thymus vulgaris*

## Abstract

**Simple Summary:**

Current knowledge of some species of *Asphondylia* (Diptera: Cecidomyiidae) is very scarce, such as those causing flower galls on Lamiaceae. Their role in natural and agricultural ecosystems remains to be investigated. Hitherto, the description of new species of this genus was mostly based on morphological variations of both the adult and the young instars. However, these variations are frequently limited and unreliable for species discrimination, and need the essential support of molecular data. Interesting aspects that are widely unexplored of this group of *Asphondylia* are their biological traits and the association with parasitoids and fungi. This paper provides an integrated description of two species of *Asphondylia* infesting flowers of *Clinopodium vulgare* and *Micromeria graeca*, and also provides data on host range and symbiosis with associated fungi.

**Abstract:**

An integrative study on some species of *Asphondylia* was carried out. Two species of gall midges from Italy, *Asphondylia rivelloi* sp. nov. and *Asphondylia micromeriae* sp. nov. (Diptera: Cecidomyiidae), causing flower galls respectively on *Clinopodium vulgare* and *Micromeria graeca* (Lamiaceae), are described and illustrated. The characteristics of each developmental stage and induced galls are described, which allowed the discrimination of these new species in the complex of *Asphondylia* developing on Lamiaceae plants. Molecular data based on sequencing both nuclear (ITS2 and 28S-D2) and mitochondrial (COI) genes are also provided in support of this discrimination. Phylogeny based on nuclear markers is consistent with the new species, whereas COI phylogeny suggests introgression occurring between the two species. However, these species can also be easily identified using a morphological approach. Phenology of host plants and gall midges are described, and some peculiar characteristics allow the complete and confident discrimination and revision of the treated species. Gall-associated fungi were identified as *Botryosphaeria dothidea,*
*Alternaria* spp., and *Cladosporium* spp.

## 1. Introduction

Gall midges (Cecidomyiidae) are one of the largest and most diverse families of Diptera, with about 6600 described species worldwide. Among the six subfamilies belonging to this family, the subfamily Cecidomyiinae is the largest and most diverse, and includes fungivorous, herbivorous, and predatory species. The currently accepted classification of the Cecidomyiinae is morphology based, but phylogenetic analysis has corroborated this classification [1,2].

The genus *Asphondylia* Loew (Diptera: Cecidomyiidae) belongs to the Asphondyliini tribe, subtribe Asphondyiliina, the latter of which is a monophyletic group including about 300 species of gall makers [1,3]. A secondary association with fungi in the form of “ambrosia galls” evolved in the clade Asphondyliini, but there is no evidence that these symbioses lead to benefits in terms of a wider host range and faster species diversification [1].

Among the 20 *Asphondylia* species recorded on Lamiaceae plants, 18 are from the Palaearctic region [3,4]. A study project on the *Asphondylia* species, causing flower galls on Lamiaceae, started in 2014 and several papers have been published on this topic [4,5,6,7].

Among the species reared from Lamiaceae, *Asphondylia serpylli* Kieffer and *Asphondylia hornigi* Wachtl cause galls on two medicinal plants consumed by humans, respectively on *Thymus vulgaris* L. and *Origanum vulgare* L. These species have recently been studied and partly redescribed, also providing several biological data [5,6].

Many of the species developing on Lamiaceae have been described and characterized only morphologically, through subtle and sometimes inconsistent differences. Based on the assumption that these species are monophagous, several Asphondyliini have been considered as new species, mainly because they were collected on different hosts following the “different host equal new species” rule.

In recent years, the integrative approach that considers several lines of evidence (morphological, molecular, and biological data) has shown the likely existence of cryptic species and the polyphagy of some of these species. Based on this new knowledge, extending this approach to other entities reared from different host plants is necessary [4,8,9,10,11,12,13].

In the present paper, two new species of *Asphondylia*, causing flower bud galls respectively on *Clinopodium vulgare* L. sensu lato (s.l.) and some subspecies of *Micromeria graeca* (L.) Benth. ex Rchb. subsp. *graeca*, *M. g.* subsp. *fruticulosa* (Bertol.) Guinea, and *M. g.* subsp. *tenuifolia* (Ten.) Nyman (Lamiaceae), are described from Italy and biological notes are given.

It is commonly accepted that larvae of Asphondyliini feed on a mycelium developing inside the galls, establishing a fundamental symbiotic relationship with the widespread endophytic fungus *Botryosphaeria dothidea* (Moug. ex Fr.) Ces. and De Not., although no conclusive demonstration has been obtained so far [14,15]. Hence, data concerning the fungal species recovered as gall associates of the new *Asphondylia* species are also presented.

The parasitoids associated with the mentioned *Asphondylia* spp. have been recorded in a previous paper [7].

## 2. Materials and Methods

### 2.1. Insect Sampling and Morphological Characterization

From June 2017 to January 2019, random samples of stems with flowers of *Clinopodium vulgare* (Figure 1a; nomenclature of the vascular plants mentioned in the text follows [16]) were collected in several locations of Basilicata and Campania regions (Southern Italy) and periods of the year.

Monthly sampling was carried out from spring to autumn in Rivello (Basilicata). Samples of about 20 stems with flowers of *M. graeca* subspp. (Figure 1b) were cut with shears and collected in several locations in Italian regions (Basilicata, Campania, Latium, Tuscany) from April 2016 to January 2018, in the first week of the month. A sample of stems with flowers of *Micromeria graeca* subsp. *consentina* (Ten.) Guinea was collected in Sersale (Calabria). A fortnightly sampling was made in Portici (Campania) to study the unknown phenology of the gall midge. Some specimens of *Asphondylia* spp. were collected on *Origanum vulgare* L. s.l., *Thymus vulgaris* L. subsp. *vulgaris*, and *Clinopodium nepeta* (L.) Kuntze also in Albania, Croatia, and Poland. Specimens collected on *Clinopodium nepeta* (L.) Kuntze s.l. and *Clinopodium menthifolium* (host) Merino subsp. *menthifolium* complement those used in the previous study [4]. All analyzed specimens are listed in Table 1.

The sampled material was transferred to the laboratory and maintained in plastic bags or plastic boxes at room temperature and natural photoperiod (from November to April: 20–22 °C and 10:14 Light:Dark (L:D) h; from May to October: 23–25 °C and 16:8 (L:D) h). Dissection of flower buds, completed with the Olympus BX51 microscope (Olympus Corporation, SZX-16, Tokyo, Japan), allowed the collection of the young stages of the gall midges, to observe their development and behavior. Adults, larvae, and pupae were preserved in 70% ethanol (Merck KGaA, Darmstadt, Germany). Specimens for the molecular analysis were randomly chosen among individuals reared from different host species using a maximum of two individuals per location and sampling date.

The adults chosen for molecular analysis were singularly placed in vials containing 95% ethanol, and preserved at −20 °C until use. In total, 76 adults, 25 larvae, and 20 pupae were slide-mounted using Balsam-phenol (Merck KGaA, Darmstadt, Germany) as a permanent medium. Mounted specimens were examined and measured under a Zeiss Axiophot microscope (Carl Zeiss, Oberkochen, Germany) was used with phase contrast and pictures were taken by a Canon Powershot 545 (Ōta, Tokyo, Japan). The terminology of adults, pupae, and larvae follows [17]. All data are presented as mean values with standard deviation (±SD).

### 2.2. Molecular Characterization of Gall Midge

Total genomic DNA extraction from whole single specimens listed in Table 1 was performed as reported in [18]. The expansion segment D2 of the 28S ribosomal subunit (28S-D2) along with the internal transcribed spacer 2 (ITS2), and the mitochondrial cytochrome c oxidase subunit I (COI) were amplified, sequenced, assembled, and edited following the protocols and methodologies reported in [4]. Generated sequences were deposited in GenBank with accession numbers reported in Table 1. Only one of the multiple identical DNA sequences was used for phylogenetic analysis. Equal sequences are summarized in Table 1. ITS2 alignment, due to many insertions and deletions, was processed using Fastgap [19], which allowed for coding these mutations as traits for Bayesian analysis. Phylogenies were reconstructed using Bayesian inference (BI) using MrBayes 3.2 [20] on COI alignment, nuclear genes alignment (ITS2+28S-D2), and three combined markers (COI+ITS2+28S-D2) alignment. The TIM1+G and F81 evolutionary models selected by jModeltest [21] were used for the COI and nuclear markers alignments, respectively. Using PartitionFinder [22] on the five partitions (one for each codon position of COI, one for nuclear genes, and one for the simple-coded gap characters from Fastgap (SCG)), evolutionary models for each alignment were selected as following: GTR+G, HKY, GTR, F81, and GTR+G. For BI, two parallel runs of four simultaneous Monte Carlo Markov chains were run for 1, 1 and 5 million generations for COI, ITS2+28-D2, and COI+ITS2+28S-D2+SCG alignments respectively; trees were sampled every 1000 generations with a burnin value set at 25%.

Sequences of *A. nepetae* [4] were added to alignments, while *Asphondylia pruniperda* Rondani 28S-D2 and COI sequences (Genbank accession number MG684646-MG684840 respectively) were used as an outgroup to root the COI and combined data trees.

In addition, MEGA6 software [23] was employed to calculate variable and parsimony informative sites of nuclear markers between and among groups. Taxa were grouped based on preliminary morphological and phylogenetic analyses.

### 2.3. Fungi Associated with Gall Development

Isolations from the gall walls and the body surface of larvae and pupae of the gall midges and their parasitoids, from samples collected in the several locations, were carried out on potato-dextrose agar (PDA, Oxoid) amended with 85% lactic acid (1 mL L^−1^) in 90 mm diameter Petri dishes (Falcon®, Becton, Dickinsinon Oxford, UK). With a comparative intent, isolations were also carried out from the ovary and the receptacle of normal flowers following the same procedure. Fragments from gall walls or the inner flower constituents were cut and transferred on the agar medium by using pins previously sterilized in 96% ethanol. Preliminary surface sterilization of the galls was not considered, since they are mostly green and soft throughout the plant vegetative period, and the use of sterilizing agents could have affected the isolation outcomes. Insect larvae and pupae were placed onto the agar medium without preliminary dissection. Plates were incubated in darkness at 25 °C. Hyphal tips from the emerging fungal colonies were transferred to fresh PDA plates (HiMedia, Mumbai, India) for morphological identification and storage of pure cultures. The preliminary ascription to genera or morphotypes was assessed through light microscopy (Olympus Corporation, BX51, Tokyo, Japan)). Sporulation by isolates of the botryosphaeriaceous morphotype was induced in cultures prepared in plates containing 2% water agar (WA) topped with sterilized pine needles, which were kept at room temperature under near-UV illumination [24]. A more circumstantial identification of selected strains was performed through rDNA-ITS sequencing. In this regard, total genomic DNA was extracted from fresh mycelium taken from pure cultures, as described in [25]. The concentration and purity of DNA samples were assessed by measuring the absorbance with Varioskan Flash (Thermo Fisher Scientific, Waltham, MS, USA). PCR amplification was carried out with the specific primers ITS1-F [26] and ITS4 [27]. Cycling parameters consisted of 40 cycles of denaturation at 94 °C for 1 min, annealing at 53 °C for 45 s, and elongation at 72 °C for 1 min. To ensure good quality sequences over the entire length of the amplicons, the forward and reverse sequences were aligned with MUSCLE [28], and only overlapping regions were used as a query for BLASTn searches in the NCBI nr/nt database. DNA extraction was also performed directly from the galls by the same method. All rDNA-ITS sequences obtained in the present study have been deposited in GenBank.

## 3. Results

The specimens of *Asphondylia* collected from *C. vulgare* and *M. graeca* showed some morphological, molecular, and bio-ecological traits that allow distinguishing them from the allied species known on Lamiaceae. Hence, they are here described as two new species. In particular, the combined phylogenetic reconstruction (COI+ITS2+28S-D2+SCG) resulted in two principal clades (Figure 2). The first clade grouped samples of *A. hornigi* and *A. serpylli* without any distinction related to the host plant. The second clade included *A. nepetae* as the sister group of all *Asphondylia* specimens reared from *Micromeria* spp. And a monophyletic group comprising all specimens reared from *C. vulgare*.

Nuclear markers alignment highlighted the absence of a 30 bps indel in samples of *A. nepetae*, both from *Clinopodium nepeta* s.l. and from *C. menthifolium* subsp. *menthifolium* (Table 1 and Appendix A). MEGA 6 software, treating gaps as missing characters, highlighted only eight variables and five parsimony informative characters in nuclear markers alignment. The nuclear markers phylogeny reconstruction confirmed the four groups obtained with the combined analysis (Appendix A), although the resulting phylogenetic relationships were diverse: specimens of *A. nepetae* and specimens reared from *Micromeria* spp. were confirmed as the sister group although the high BI posterior probabilities support the discrimination between them, while *A. hornigi* and *A. serpylli* grouped together as the sister group of specimens reared from *C. vulgare*. This last phylogenetic relationship is also confirmed in COI phylogeny (Appendix A), but, in turn, *A. nepetae* and specimens from *Micromeria* spp. grouped together due to the sharing of several Mt haplotypes (Table 1). In this group, in particular, eight mitochondrial haplotypes have been found on 31 samples, and 48% of specimens had haplotype A which is shared with the majority of *A. nepetae* sampled by Bernardo et al. [4].

### 3.1. Asphondylia rivelloi sp. nov. Viggiani

#### 3.1.1. Morphological Characterization

##### Adult

**Female**. Body length: 1.8–2.0 mm. Palpus 3-segmented, first segment about half the length of the second, third segment 1.5–2.3× longer than the second (Figure 1c). Antenna with scape subtrapezoidal, distally enlarged, twice as long as the transverse or subquadrate pedicel (Figure 1d); flagellomeres with F1 3.7–5.5× as long as wide, F2–F10 gradually shorter, F11–F12 subglobular (Figure 1e). Seventh abdominal sternite 1.0–1.1× as long as wide; ratio length of ovipositor/length of seventh abdominal sternite in average 2.68× (±0.262; min. 2.4; max. 3.3; *n* = 12). Typical ovipositor of *Asphondylia* (Figure 1f), length of the needle part on average 1.37 ± 0.143 mm (min. 1.2; max. 1.67; *n* = 12). Wing 2.5–2.8× as long as wide, venation as in Figure 1g. Legs with tarsal claws bent beyond mid length, with empodium as long as claw (Figure 1h).

**Male.** Palpus 3-segmented, first segment transverse, second segment ovoid, about twice the length of the first, third segment elongate, 1.5–2.0× longer than the second (Figure 3a). Antenna with scape subtrapezoidal, distally enlarged, twice as long as the transverse or subquadrate pedicel (Figure 3b); flagellomeres subcylindrical, with F1 4.1–4.4× as long as wide, F2 shorter, 0.8× F2, F3–F12 gradually shorter, F12 3.2–4.4× as long as wide, distally pointed (Figure 3c). Wing is the same as in the female. Genitalia (Figure 3d) without any significant differences with *A. nepetae* ones [4].

##### Young Stages

**Last instar larva.** Length: 2.3–3 mm. Body orange, deeply segmented, tapering posteriorly (Figure 4a), sternal spatula quadridentate (Figure 4b), the inner pair of teeth usually slightly shorter than the outer. No tenable morphological character for discrimination from that of *A. nepetae*.

**Pupa.** Length: 1.6–2.8 mm. Body ovoid and brown-reddish (Figure 4c). Antennal horns rather elongated, subtrapezoidal, closely approximated at base and obliquely trunked (Figure 4e). Upper frontal horns short, bifid (Figure 4f); lower frontal horn (Figure 4g) like a small crest around a groove. Abdominal segments II–VII dorsally with a verrucose sculpture on a basal and distal band; basal half with irregularly sparse spines in 1–2 rows; distal verrucose band preceded by a regular row of longer and wider spines; last segment distally with two lateral pairs of larger spines preceded by shorter and thinner spines, increasing in size from the basal margin of the abdominal segment (Figure 4d).

**Diagnosis.** The new species is defined by a combination of the following features: female and male with labial palp 3-segmented; length of the needle part of the ovipositor 1.2–1.67 mm; antennal horns of the pupa rather elongated, subtrapezoidal, closely approximated at base, obliquely trunked; upper frontal horns short, bifid; lower frontal horn like a small crest around a groove. *Asphondlia rivelloi* differs from *A. nepetae* by having F1 < 5.6 as long as wide, and from *A. micromeriae* by having the seventh abdominal sternite < 1.2 as long as wide. Furthermore, *A. rivelloi* can be distinguished from *A. micromeriae*, *A. nepetae*, *A. hornigi*, and *A. serpylli* by different COI and ITS2 + 28S D2 sequences (Figure 2, Appendix A).

**Gall.** The flower gall is formed by the normal calyx that wraps the true gall (Figure 4h), an ovoidal body, in which the midge and the associated fungi develop. The dimensions and the shape of the flower gall containing the mature larva or the pupa are similar to those of a mature flower containing fruits (max. width: 1.8–2.5 mm) (Figure 4i,l), but the flower fruits in uninfested flower are asymmetrically convex because they have one side with a more convex profile.

**Etymology.** The specific name *rivelloi* is referring to the native village of the author, where the gall midge phenology was mostly studied, i.e., Rivello.

**Host plant.***Clinopodium vulgare* s.l.

**Distribution.** Italy, Basilicata: Lagonegro, Rivello, Tricarico; Campania: Monte Terminio, Pietraroja, Serino, Roccarainola, Roccadaspide, Tramonti-Valico di Chiunzi.

**Material examined.** Holotype♀. Italy, Basilicata, Rivello, countryside Filoto, 6.iv.2016 (laboratory emergence) from flower gall on *C. vulgare*, collected on 22.ii.2016, coll. G. Viggiani. Paratypes. 2♀, same data of holotype; 1♀, Rivello, 29.xi.2015 (laboratory emergence), same host plant and collector; 1♀, Rivello, 2.iv.2016 (laboratory emergence) from flower gall collected on 14.xii.2015, same host plant and collector; 1♂, Rivello, 9.iii.2016 (laboratory emergence) from flower gall, collected on 14.xii.2015, same host plant and collector; 1♂, Rivello, 6.iv.2016 (laboratory emergence), from flower gall, collected on 14.xii.2015, same host plant and collector.

Holotype and paratypes are deposited in the entomological collection of the Department of Agriculture, University of Naples ‘Federico II’, Portici, Italy.

**Comment.** Fedotova [29] described the species *Asphondylia clinopodiiflorae* from Russia, causing flower galls on *C. vulgare*. This species is characterized by 2-segmented labial palps in the male, which are 3-segmented in the female; indistinct apical prominences of the antenna in the pupa; upper frontal spine in the form of two wide connected zigzag without wrinkles at the sides; and lower frontal spine absent. The new species *A. rivelloi*, causing flower galls on the same host plant, differs from *A. clinopodiiflorae* in having 3-segmented labial palps in the male, antennal horns of the pupae are prominent, elongated, subtrapezoidal, and lower frontal horn is like a small crest around a groove. The longer ovipositor distinguishes *A. rivelloi* from the allied species of *Asphondylia*, in which this character was evaluated. This is the first record of two species of *Asphondylia* causing flower galls on Lamiaceae developing on the same host plant.

**Phenology of host plant and gall midge.** The phenology of *C. vulgare* was studied continuously since 2015, with an interval of two weeks, in Rivello (PZ), countryside Filoto at 410–450 m asl, but additional observations have been made irregularly in other locations in Basilicata and Campania. The flowering period starts in the late May to the beginning of June and continues until late autumn, with a break or slower trend during the hottest months (August and September). The duration of the unflowering period is also linked to the rain trend. With a rainy spring and autumn, 2018 was favorable for a long and abundant flowering period of *C. vulgare*, contrary to 2017 and 2019. All the overwintering flower galls examined in the winter (Figure 4i) contained pupae in a quiescence state that could be interrupted at a temperature above 20 °C. In fact, from pupae in flower galls that were kept from 13.i.2016 at a constant temperature of 25 °C, adults emerged 30–40 days later. The gall midge starts to oviposit in late May to the beginning of June. Young stages develop singularly in one flower gall from egg to adult, emerging from the top of the gall with the removal of the dried, distal part of the corolla (Figure 4i).

According to the data reported in Table 2, the percentage of stems of *C. vulgare* with flower galls on *A. rivelloi* varied from 3.3 to 35%.

### 3.2. Asphondylia micromeriae sp. nov. Viggiani

#### 3.2.1. Morphological Characterization

##### Adult

**Female.** Body length: 1.5–1.8 mm. Antennae brown, eyes black, palpi grey, thorax, abdomen, and legs dark brown. Eye facets close together, hexagonoid, eye bridge 8 to 10 facets long medially (Figure 5a). Palpus is 3-segmented, first segment as long as wide, about half of the length of the second, third segment 1.5–2× longer than the second. Antenna with scape is twice as long as the transverse pedicel (Figure 5b), flagellomeres (Figure 5c) with the following length/maximum width ratio (*n* = 10): F1 (18.5/4.1), F2 (15.5/4.3), F3 (14/4.4), F4 (13.2/4.2), F5 (13.4/4.2), F6 (12.7/4.1), F7 (11.8/4.1), F8 (9.3/4.5), F9 (7.4/4.5), F10 (4.9/4.9), F11 (4.8/5.1), F12 (3.4/4.1). Seventh abdominal sternite 1.2–1.4× as long as wide, and 2.0–2.5× the length of the sixth; ratio length of ovipositor/length of seventh abdominal sternite on average 2.16× (± 0.100; min. 2.0, max. 2.4; *n* = 22). Typical ovipositor of *Asphondylia* (Figure 5d), length of the needle part on average 0.86 ± 0.075 mm (min. 0.70, max. 0.99 mm; *n* = 20). Wing 2.4–2.6× (average on *n* = 7) as long as wide, with R_5_ vein ending near wing apex and other characters, as in Figure 5e. Legs with ventrodistal spine of the foreleg first tarsomere one-eighth of the latter length, bent at distal half. Claws simple, slightly longer than empodium.

**Male.** Body length: 1.4–1.8 mm. Antenna with scape subtrapezoidal, distally enlarged, twice as long as the transverse pedicel (Figure 5f); flagellomeres cylindrical, with dense circumfila; first flagellomere 3.6–4.5× as long as wide (*n* = 7), the subsequent gradually shorter, distal two flagellomeres (11–12) respectively 2.2–3.0× and 2.2–2.7× (*n* = 7) as long as wide (Figure 5g). Genitalia showing the typical shape of other *Asphondylia* (Figure 5h); genital capsule (*n* = 5) slightly wider than long (average: 3.9:3.3); gonocoxite 2.0–2.5× (*n* = 10) as long as gonostyle, with subtriangular and setose lobes, twice as wide as long, and with long setae on the distal margin; gonostyles (*n* = 10) ovoidal, 1.2–1.3× as long as wide, with two apical short and sclerotized teeth; cerci distally divided into two triangular and setose lobes; aedeagus with a large base followed by a distally pointed rod about 9× as long as wide.

##### Young Stages

**Egg.** (Figure 5i), first (Figure 6a), and second instar larva without any significant difference in shape and characters with those of *A. nepetae* [4].

*Last instar larva*. Similar to *A. nepetae* ones (Figure 6b), but smaller (length: average 1.54 ± 0.211 mm; min. 1.1; max. 1.8; *n* = 10), with a shorter spatula (total length average: 0.16 mm ± 0.012; min. 0.13, max. 0.18 mm; *n* = 15) (in *A. nepetae*, total length average: 0.19 ± 0.008; mm; min. 0.17, max. 0.21 mm; *n* = 15).

**Pupa.** Indistinguishable from *A. nepetae* pupa [4], but smaller (length: average 1.80 ± 0.169 mm; min. 1.6, max 2.0 mm; *n* = 10), the last three segments dorsally with spines as in Figure 6g, spatula quadridentate (Figure 6c), antennal horns (Figure 6d) distally slightly longer than wide, and the lower frontal horns (Figure 6f) frequently inconspicuous.

**Diagnosis.** The new species is defined by a combination of the following features: length of the needle part of the ovipositor 0.70–0.99 mm; antennal horns of the pupa distally slightly longer than wide, and the lower frontal horns frequently inconspicuous. *Asphondylia micromeriae* differs from *A. rivelloi* by having the seventh abdominal sternite 1.2–1.4 as long as wide, ratio length of ovipositor/length of the seventh abdominal sternite in average ≤ 2.4, and from *A. nepetae* by having all the flagellomeres longer and narrower (Appendix A). This species can be distinguished from *A. rivelloi*, *A. hornigi*, and *A. serpylli* by having different COI and ITS2+28S D2 sequences (Figure 2, Appendix A), and from *A. nepetae* by having different ITS2+28S D2 sequences (Appendix A).

**Gall.** The flower transformed in a full gall (Figure 6i,m) rather normally, with a symmetrical shape, slightly wider (average width: 1.5 ± 0.12 mm; min. 1.3, max. 1.7 mm; *n* = 15) than the flower containing fruits (average width: 1.0 ± 0.07 mm; min. 0.9, max. 1.2 mm; *n* = 15) (Figure 6h). As in other species (*A. hornigi*, *A. nepetae*, *A. serpylli*) [4,5,6], the overwintering gall shows a dried calyx (Figure 6k) covering an ovoidal hard body, the true gall (Figure 6l), which is like a seed with the gall wall internally covered with a thick and compact black layer of fungal mycelium.

**Etymology.** The specific name *micromeriae* refers to the host plant genus, i.e., *Micromeria*.

**Host plants.***Micromeria graeca* subsp. *graeca*, *M. graeca* subsp. *fruticulosa*, *M. graeca* subsp. *tenuifolia*.

**Distribution.** Italy, Basilicata: Pietrapertosa, Rivello, Trecchina; Campania: Capo d’Orso, Corbara, Felitto, Roccadaspide, Orria-Santoianni, Pago di Vallo della Lucania, Pisciotta, Ravello, Salerno, Scala-Pontone, Torraca, Bacoli, Boscotrecase, Capri, Napoli, Palma Campania, Portici, Pozzuoli, Pozzuoli-Astroni, Pozzuoli-Lucrino, Procida, Vico Equense, Vivara, Bagnoli Irpino; Lazio: Minturno; Toscana: Cortona.

**Material examined.** Italy. Holotype ♀, Portici, 1.x.2016 (laboratory emergence) from flower gall on *Micromeria graeca* subsp. *graeca*, collected on 30.ix.2016, coll. G. Viggiani. Paratypes. 1♀, Orria-Santoianni, 14.x.2016 (laboratory emergence) from the same host of the holotype collected on 4.x.2016, coll. F. Nugnes; 1♀, Scala-Pontone, 27.iv.2016 (laboratory emergence) from the same host of the holotype collected on 9.iv.2016, coll. R. Nicoletti; 1♀, Portici, 16.iv. 2017, same data of the holotype coll. F. Nugnes; 4♀, Minturno, 5–10.v.2017 (laboratory emergence) from the same host of holotype collected on 1.v.2017, coll. F. Nugnes; 1♀, Pisciotta, 20.x.2016, from the same host of holotype, coll. F. Nugnes; 4♂, Capri, 21.iv.2017 (laboratory emergence) from the same host of holotype collected on 9.iv.2017, coll. R. Nicoletti; 2♂, Pozzuoli-Astroni, 3.iv.2017 (laboratory emergence) from the same host of holotype collected on 27.i.2017, coll. R. Nicoletti; 1♂, Portici, 21.iv.2016 (laboratory emergence) from the same host of holotype collected on 14.iv.2017, coll. G. Viggiani. Holotype and paratypes are deposited in the entomological collection of the Department of Agriculture, University of Naples ‘Federico II’, Portici, Italy.

**Comment.** Up to now no species of *Asphondylia* has been described on *Micromeria* spp. Cerasa [30] recorded an *Asphondylia* species from Sicily on *Micromeria graeca* subsp. *fruticulosa* and gave some biological notes. This species is very probably *A. micromeriae*. The new species differs from the known allied *Asphondylia* on Lamiaceae in the smaller size, shorter ovipositor, and reproductive activity starting earlier in the year.

**Phenology of host plant and gall midge.** The new species is associated with several subspecies of *Micromeria graeca*. The genus *Micromeria* includes about 54 accepted species with 32 subspecies and 13 varieties of perennial herbs, subshrubs or shrubs, rarely annual herbs that are more or less aromatic distributed from the Macaronesian-Mediterranean region to southern Africa, India, and China [31]. To date, in Italy, six subspecies are recognized. However, the diagnostic characters between these subspecies are not well defined and further investigations that aim to clarify the taxonomic value of these taxon are needed [32]. Hosts of *A. micromeriae* are *Micromeria graeca* subsp. *graeca*, *M. g.* subsp. *fruticulosa*, *M. g.* subsp. *tenuifolia*, and *M. graeca* subsp. *consentina*. The gall midge can reproduce from the end of winter to autumn. The first generation takes place in March–April on the flower buds of *M. g.* subsp. *fruticulosa*, in late April–June on *M. g.* subsp. *graeca*, and in June to the beginning of July on *M. g*. subsp. *tenuifolia*. Then, the generations overlap and gall midge development is linked to the flowering period of the host plant in specific environments. Flowers with eggs of *Asphondylia* have been found until September. As reported for *A. nepetae* [4], the full larvae of the last generation become quiescent from late autumn and overwinter as pupae in the flower galls. The emergence of the first adults of the year (Figure 6j) starts in late March–April.

### 3.3. Fungi Associated with Galls of A. rivelloi and A. micromeriae

As a result of fungal isolations from galls, *B. dothidea* was found to be systematically associated with the gall midges in all samples collected from *C. vulgare* and *M. graeca* subspp., confirming previous findings on other Lamiaceae and many different plant species [4,5,14,33]. Although no pycnidia were observed directly on galls, botryosphaeriaceous isolates regularly formed pycnidia on WA under UV light within one week, producing hyaline aseptate fusoid conidia corresponding in shape and size to the description of *Fusicoccum aesculi*, the anamorphic stage of *B. dothidea* [34]. Species identification was confirmed through a GenBank blast of the rDNA-ITS sequences obtained for isolates from *M. graeca* subsp. *fruticulosa* (MfCa3, MK348523), *M. graeca* subsp. *graeca* (MgVi9, MK348524), and *C. vulgare* (ClRi2, MN731272) [15], which showed 100% homology with dozens of sequences from strains of this species, including the epitype CBS115476. The same results for an ITS sequence (MK348522) were obtained after extracting DNA from a random gall collected on *M. graeca* subsp. *graeca* at the Astroni Nature Reserve. On the other hand, this fungus was never found in healthy flower buds, unlike both *Cladosporium* and *Alternaria*, which are well known for their epiphytic occurrence. Isolates ascribed to these genera were also recovered from gall fragments and inquilines from all locations, and again their occurrence was considered to derive from saprophytic growth on the dead flower tissues, which probably progresses in the time elapsing between collection and isolation (usually one day or more). The hypothesis that saprophytic fungi are recovered at a higher extent if isolation is not quickly completed after collection of plant samples was verified through a more systematic isolation plan from galls collected on *M. graeca* subsp. *graeca* at the Astroni Nature Reserve, where a mycological laboratory was available, enabling us to perform isolations within 30 min from the collection of the plant material. From this site, isolations from gall walls and the insect inquilines were carried out throughout the blooming period, starting in April and continuing until vegetation was arrested by dry weather conditions in late July. Results in terms of species assortment are reported in Table 3. Besides the dominance of *B. dothidea* with about 52% isolations, the incidence of *Cladosporium* was quite reduced as compared with other locations and our previous findings concerning *T. vulgaris* and *C. nepeta* [4,5], which can be assumed to derive from the timelier isolation procedure. Conversely, *Alternaria* was still significantly represented (about one third of total isolations). In several cases, *B. dothidea* and *Alternaria* emerged from the same gall fragment, and their hyphae grew intermingled on PDA.

## 4. Discussion and Conclusions

This integrated approach has brought some lines of evidence, which, on the one hand, lead to describe two new species, *A. rivelloi* and *A*. *micromeriae*, and on the other, to suggest the synonymization of two morphological species described more than a century ago, *A. hornigi* and *A. serpylli*.

Different from the previous analysis on *A. nepeta,* COI was not conclusive because some species clustered together (Appendix A) [4]. However, phylogenetic reconstruction also based on nuclear markers (Figure 2 and Appendix A), which usually are more conserved than COI, resulted in a tree that supported the description of two new species. After all, the limitations of a molecular approach based solely on the COI gene are well known [35,36]. Indeed, COI phylogeny reconstruction suggested recent hybridization and introgression occurring between some *Asphondylia* species. Furthermore, there are some haplotypes shared between specimens of different species. However, no double-peaks or heterozygosity were found in the ITS2+28S-D2 sequences, which could support this hypothesis.

Species that cannot be distinguished through COI (*A. nepeta* and *A. micromeriae*), however, have other lines of evidence that make them distinguishable, such as morphological and biological traits, and belong to different clades in the phylogenetic reconstruction based on combined data (Figure 2). Although the specimens were collected in the same areas and sometimes a few meters apart from two different host plants (*C. nepeta* and *M. graeca*), all the specimens collected on *C. nepeta* and *C. menthifolium* showed a long deletion in ITS2+28S-D2 sequences that is absent in those collected on *M. graeca* (Table 1).

Other authors have already shown that sometimes, due to a sharing of the COI sequences, the mitochondrial DNA portion does not allow the distinction of some species which are otherwise corroborated by other markers [35,36,37,38,39].Conversely, specimens collected from oregano and thyme in several European localities do not present any distinctive morphological character and present very similar COI, with some shared haplotypes; moreover, ITS2+28S-D2 sequences are equal. These results suggest revising the morphological characters of the species *A. hornigi* and *A. serpylli* to evaluate their probable synonymy.

The description of two new species in the present paper confirms the marked homomorphy among the species that cause flower galls on Lamiaceae. Some observed variations at species-level, such as those concerning the length of the ovipositor, require a more extended investigation in several described species. These variations are probably linked to the shape and the dimensions of the flower of the infested host plant, and not only the shift on different parts of a single host plant, as found in the *Asphondylia auripila* species group [40,41]. The morphological differences of most *Asphondylia* infesting Lamiaceae, reported by the authors, concern male genitalia, full larva, and pupa. In several cases, they appear as slide artifacts or such variable characters, untenable for valid species discrimination. This situation also affects the type designation, mostly based on the male, although morphological differences concern larval and pupal characteristics. It seems reasonable to designate as a holotype given the adult or young stage in which the discriminatory characteristics of a newly described species are based, or in other cases when the taxon is represented by several stages, a hapantotype [42]. On the other hand, the flower galls produced by these species, taken as their “extended phenotype” [13], do not offer discriminatory characters. The *Asphondylia* species reproducing in flowers of Lamiaceae cause homomorphic and rather cryptic galls. The lack of any key to species identification, even for a small group of species, proves the weakness of the present taxonomic scheme. The modern trend of an integrative description of new species, based on morphological, biological, and molecular data, can help in the effort to keep the species discrimination on a more tenable basis and their comparison feasible, independent from the host plant.

Several fungi were found to be associated with the galls and midge larval development, with *B. dothidea* being the most common. The taxonomic identification was further confirmed in the course of a dedicated phylogenetic study based on strains collected on various Lamiaceae species [15]. The evidence that *B. dothidea* and *Alternaria* commonly co-occur in galls leads to a recommendation for researchers involved in investigations on galls formed by Asphondyliinae on whatever plant species to be cautious in their assessments. In fact, *B. dothidea* is known to have a *Dichomera* synanamorph producing muriform conidia in pycnidia [34], and a preconceived observer could eventually be misled by the finding of this kind of phaeodictyospores in offhand glass slides.

In this respect, conclusive proof would be obtained by checking if these conidia come out from pycnidia, in contrast to the free catenulate conidia produced by *Alternaria*-like species. Indeed, the pleomorphism characterizing *B. dothidea* is quite puzzling; while *F. aesculi* represents the most common anamorph, the *Dichomera* stage has been frequently reported in *Asphondylia* galls, and sometimes the formation of both kinds of conidia in the same pycnidium has been observed [43]. As a matter of fact, in our observations on isolates from *M. graeca* subspp. carried out up to 3 weeks from the preparation of the WA cultures, all pycnidia exclusively produced conidia of the *Fusicoccum* type. *Alternaria*-like fungi have been diffusely recovered from galls of Asphondyliinae [33,43,44,45], even if reports concerning the finding of muriform conidia in mycangia are questionable due to possible confusion with *Dichomera*. So far, identification at the species level of *Alternaria*-like isolates from cecidomyiid galls has never been approached, probably due to the quite complex taxonomy of these fungi. As for our *Micromeria* isolates, their results were found to be quite heterogeneous in terms of culture morphology, which can be indicative of species diversity.

The time-lapse between collection and isolation might explain the identification of *Cladosporium* spp. as the basic fungal associate of *Asphondylia* in pioneering reports [46,47]. In fact, *Cladosporium* spp. Represent quite a constant finding in investigations concerning cecidomyid-associated fungi [33,48,49]; moreover, the presence of *Cladosporium* conidia in mycangia has been reported, as well as isolation from adults [43], which could imply an active role by the midges in spreading these fungi. However, unlike gall isolates of *Botryosphaeria*, which invariably belong to the species *B. dothidea* regardless of the host plant, *Cladosporium* isolates display taxonomic heterogeneity. In fact, a dedicated investigation involving a sample of strains recovered as gall associates on several species of Lamiaceae showed them to belong to nine species from the *Cladosporium herbarum* and *Cladosporium cladosporioides* species complexes, including the novel species *Cladosporium polonicum*. Concerning the plant species considered in the present paper, this work reported the identification of *Cladosporium crousii* and *Cladosporium pseudocladosporioides* from *C. vulgare*, *Cladosporium europaeum* from *M. graeca* subsp. *graeca*, and *C. cladosporioides* from *M. graeca* subsp. *graeca* and *fruticulosa* [50].

## Figures and Tables

**Figure 1 insects-12-00958-f001:**
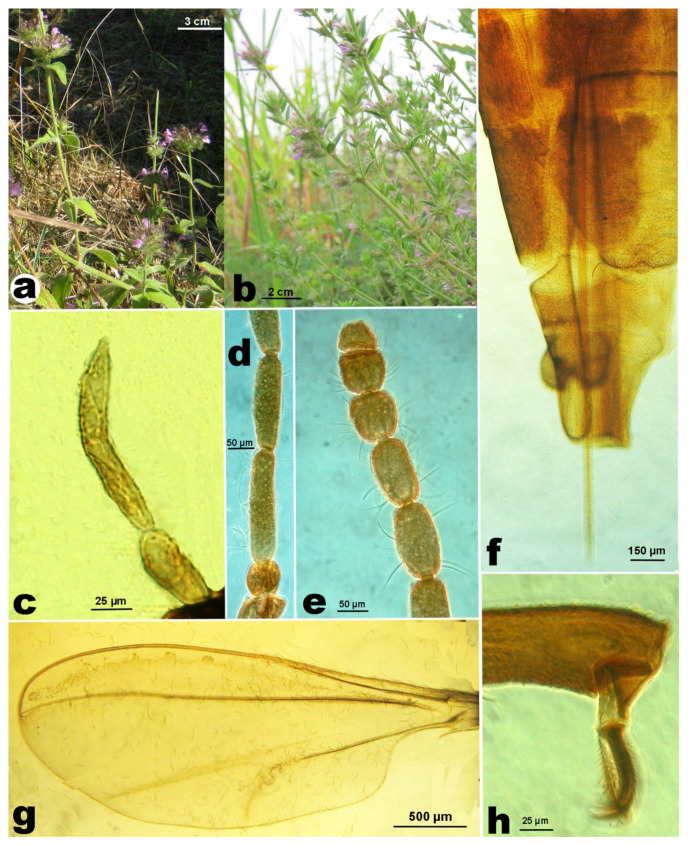
(**a**). Stems of *Clinopodium vulgare* with flowers. (**b**). Stems of *Micromeria graeca* subsp. *graeca* with flowers. (**c**). *Asphondylia rivelloi*, female, palpus. (**d**). Basal segments of antenna. (**e**). Distal segments of antenna. (**f**). Abdomen with ovipositor. (**g**). Wing. (**h**). Empodium and claws.

**Figure 2 insects-12-00958-f002:**
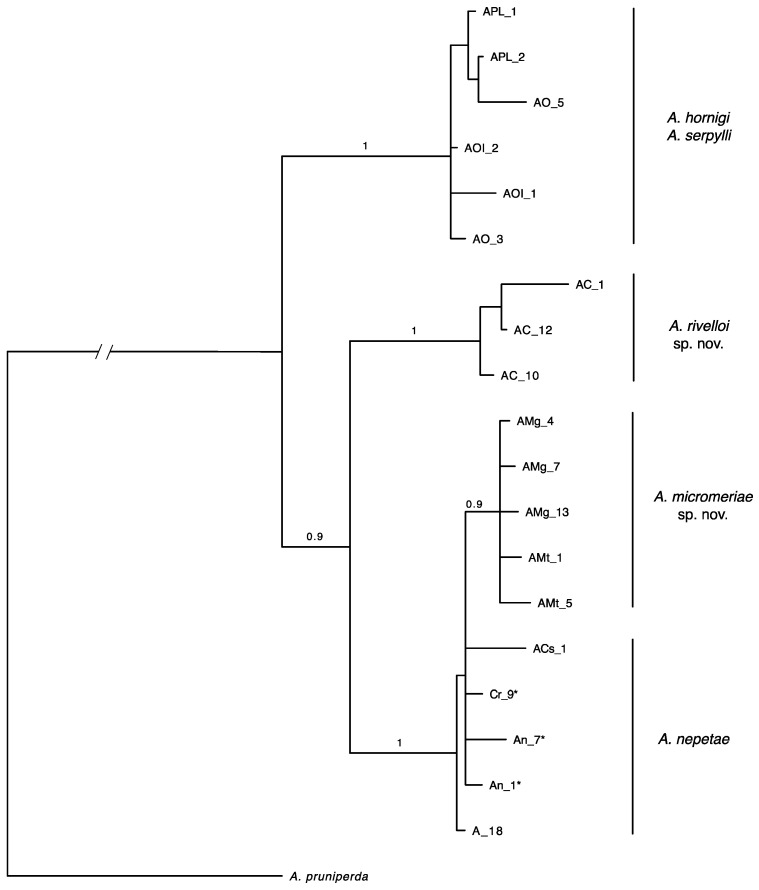
Bayesian majority rule consensus tree based on combined molecular markers (COI-ITS2+28S-D2) and binary data (SCG). Posterior probabilities ≥0.9 are shown above branches. (*) Sequences from [4].

**Figure 3 insects-12-00958-f003:**
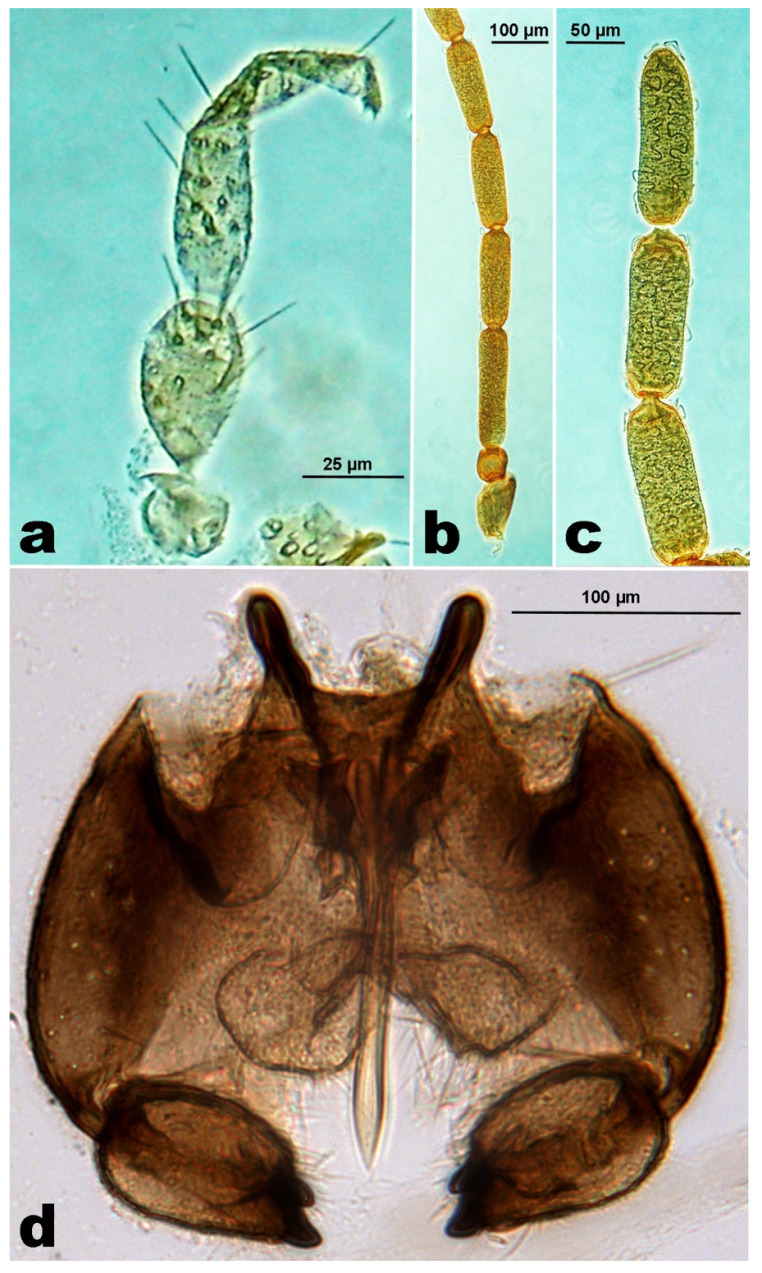
Male of *Asphondylia rivelloi*, (**a**). Palpus. (**b**). Basal segments of antenna. (**c**). Distal segments of antenna. (**d**). Genitalia.

**Figure 4 insects-12-00958-f004:**
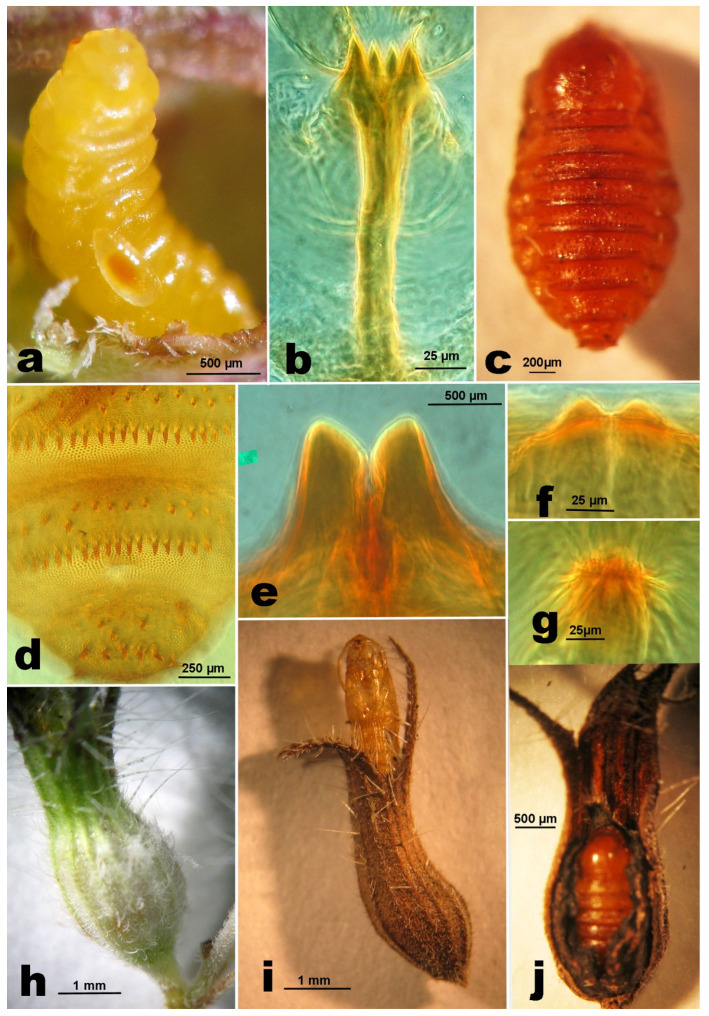
*Asphondylia rivelloi*. (**a**). Last instar larva, on it is a visible larva of an ectoparasitoid. (**b**). Spatula of the same. (**c**). Pupa. (**d**). Distal abdominal segments of the same, dorsal view. (**e**). Antennal horns. (**f**). Upper frontal horns. (**g**). Lower frontal horns. (**h**). Flower gall. (**i**). Flower gall with the pupal case of the emerged midge. (**j**). Flower gall opened showing an overwintering pupa.

**Figure 5 insects-12-00958-f005:**
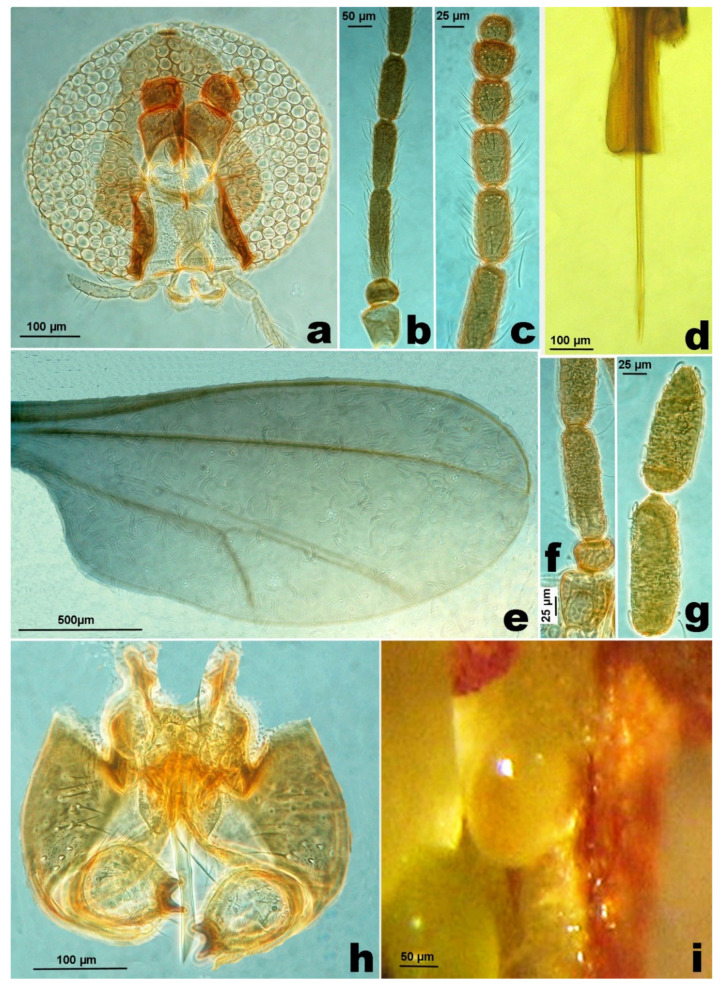
*Asphondylia micromeriae*. (**a**). Female, head, frontal view. (**b**). Basal segments of female antenna. (**c**). Distal segments of female antenna. (**d**). Ovipositor. (**e**). Fore wing. (**f**). Basal segments of male antenna. (**g**). Distal segments of male antenna. (**h**). Genitalia. (**i**). Laid *Asphondylia* egg.

**Figure 6 insects-12-00958-f006:**
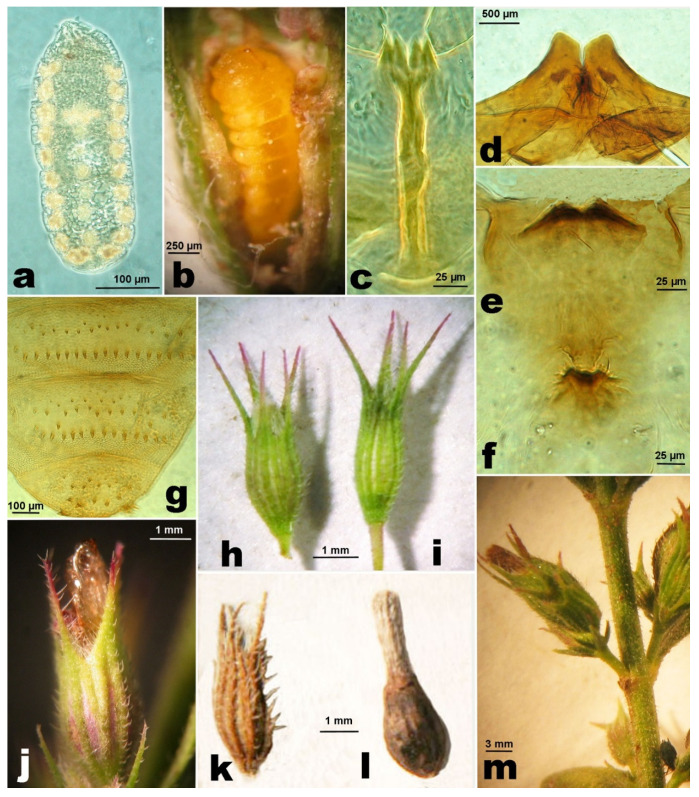
*Asphondylia micromeriae*. (**a**). First instar larva. (**b**). Last instar larva in a gall. (**c**). Spatula of the same. (**d**). Pupa, antennal horns. (**e**). Pupa, upper frontal horns. (**f**). Pupa, lower frontal horns. (**g**). Pupa dorsal last segments. (**h**). Uninfested flower. (**i**). Infested flower. (**j**). Flower gall with the pupal case of the emerged midge. (**k**). Calyx of a flower gall. (**l**). True gall. (**m**). Stem with flower galls.

**Table 1 insects-12-00958-t001:** List of specimens used for the molecular study. It: Italy; Pl: Poland; Cr: Croatia; Al: Albania; MtH: mitochondrial haplotype; +, − indicate presence or absence of the ITS2 indel, respectively.

Code.	*Asphondylia* Species	Host	Locality	MtH	ITS2 Indel	GenBank Accession Code	
						COI	ITS2-28S-D2
A18	*A. nepetae*	*Clinopodium nepeta* s.l.	Blue Eye, Al	B	−	OK047088	OK050118
A19				B	−	OK047089	OK050119
A20			Ksamil, Al	A	−	OK047090	OK050120
A22			Astroni-Pozzuoli, It	A	−	OK047091	OK050121
A23			Lucrino, It	A	−	OK047092	OK050122
A24				A	−	OK047093	OK050123
ACS1		*Clinopodium menthifolium* subsp. *menthifolium*	Rivello, It	C	−	OK047094	OK050124-OK050112
ACS2				A	−	OK047095	OK050125-OK050113
ACS4				A	−	OK047096	OK050126-OK050114
AC1	*A. rivelloi* sp. nov.	*Clinopodium vulgare* s.l.	Rivello, It	D	+	OK047097	OK050127-OK050115
AC2				D	+	OK047098	OK050128-OK050116
AC3				D	+	OK047099	OK050129-OK050117
AC10			Roccarainola, It	E	+	OK047100	OK050130
AC11				E	+	OK047101	OK050131
AC12			Serino, It	F	+	OK047102	OK050132
AC13				F	+	OK047103	OK050133
AMg1	*A. micromeriae* sp. nov.	*Micromeria graeca* subsp. *graeca*	Scala, It	A	+	OK047104	OK050134
AMg2			Palma Campania, It	A	+	OK047105	OK050135
AMg3				A	+	OK047106	OK050136
AMg4				G	+	OK047107	OK050137
AMg5				G	+	OK047108	OK050138
AMg6				G	+	OK047109	OK050139
AMg7			Lucrino, It	H	+	OK047110	OK050140
AMg8				H	+	OK047111	OK050141
AMg9			Slatine, Cr	A	+	OK047112	OK050142
AMg10				A	+	OK047113	OK050143
AMg11			Portici, It	A	+	OK047114	OK050144
AMg12			Orria, It	I	+	OK047115	OK050145
AMg13				I	+	OK047116	OK050146
AMg17			Portici, It	A	+	OK047117	OK050147
AMg18			Astroni-Pozzuoli, It	G	+	OK047118	OK050148
AMg19			Capri, It	G	+	OK047119	OK050149
AMg20				G	+	OK047120	OK050150
AMf1		*M. graeca* subsp. *fruticulosa*	Scala, It	A	+	OK047121	OK050151
AMf2				A	+	OK047122	OK050152
AMt1		*M. graeca* subsp. *tenuifolia*	Rivello, It	J	+	OK047123	OK050153
AMt2				G	+	OK047124	OK050154
AMt5				K	+	OK047125	OK050155
AOR1	*A. hornigi*	*Origanum vulgare* s.l.	Fajstawice, Pl	L	+	OK047126	OK050156
AOR2				L	+	OK047127	OK050157
AOP1			Lublin, Pl	L	+	OK047128	OK050158
AOP2				L	+	OK047129	OK050159
AOI1			Castellammare di Stabia, It	M	+	OK047130	OK050160
AOI2				N	+	OK047131	OK050161
AO3			Blue Eye, Al	O	+	OK047132	OK050162
AO5				P	+	OK047133	OK050163
APL1	*A. serpylli*	*Thymus vulgaris* subsp. *vulgaris*	Lublin, Pl	L	+	OK047134	OK050164
APL2				L	+	OK047135	OK050165
APL3				L	+	OK047136	OK050166
APL4				L	+	OK047137	OK050167
APL5				L	+	OK047138	OK050168
APL6				L	+	OK047139	OK050169
APL7				L	+	OK047140	OK050170
APL8				L	+	OK047141	OK050171
ATh1			Boniewo, Pl	L	+	OK047142	OK050172
ATh2				L	+	OK047143	OK050173

**Table 2 insects-12-00958-t002:** Percentage of stems with flower galls on *A. rivelloi*.

Sampling Date	Locality	Nr. of Examined Stems	% of Stems with Galls
28.vi.2017	Rivello	20	15.0
17.vii.2017	Rivello	20	10.0
03.ix.2017	Rivello	24	25.0
21.ix.2017	Serino	20	35.0
10.x.2017	Rivello	37	8.0
25.vi.2018	Rivello	50	16.0
30.vi.2018	Monte Terminio	45	6.6
01.vii.2018	Tramonti-Valico di Chiunzi	23	13.0
09.vii.2018	Rivello	28	3.5
04.viii.2018	Pietraroja	40	7.5
03.ix.2018	Rivello	30	0
13.ix.2018	Roccadaspide	30	3.3
24.ix.2018	Serino	10	10.0
15.xii.2018	Rivello	110	9.0
2.i.2019	Rivello	50	12.0

**Table 3 insects-12-00958-t003:** Gall-associated fungi isolated from samples collected on *Micromeria graeca* at the Astroni Nature Reserve.

Fungus	Number of Isolates		
	Gall Walls	Larvae	Total
*Botryosphaeria dothidea*	51	17	68
*Alternaria*-like	34	10	44
*Cladosporium* spp.	10	6	16
*Penicillium* sp.	-	1	1
Unidentified	1	1	2
Total	96	35	131

## Data Availability

Sequences were deposited in GenBank. All other data presented in this study are available on request from the corresponding author.

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
