# Peer review of "An Integrative Study on Asphondylia spp. (Diptera: Cecidomyiidae), Causing Flower Galls on Lamiaceae, with Description, Phenology, and Associated Fungi of Two New Species"

_insects, 2021, doi:10.3390/insects12110958_

Round 1

Reviewer 1 Report

The manuscript contains a small number of formal and stylistic errors – e.g. the species names are often not in italics (L64, L321, L475 and elsewhere in the text) or short dash is used instead of long hyphen (L184, L187, L201 and repeatedly in the text). Polishing would be beneficial for the text.

L136-L138: „Preliminary surface sterilization of the galls was not considered since they are mostly green and soft throughout the plant vegetative period, and the use of sterilizing agents could have affected the isolation outcomes“. The isolation from uninfested parts of the plant is certainly a good control method, although control in the form of a gall surface would also be interesting. I have no experience with isolating of fungi from galls such tiny as the galls of so-called „A. serpylli“ or „A. hornigi“. Do the author think that the gentle surface sterilization with PBS and Tween oil, standardly used for example in research of fungal symbionts of ambrosia beetles, is really inferior method also for soft green galls?

L201, L305: Based on experience with other Cecidomyiidae, the length of the body of larvae and pupae is quite conditioned by the environment (specific host plant, etc.) and thus is not a good determinant. It is also stated that n = 10 larval individuals were measured (or n = 15). A more important number would be the number of populations from which the larvae were extracted, as otherwise the number of individuals within one population is pseudoreplication. If there were only a few localities of sampling for this measurement, the number may not indicate real differences among species.

Figure 4a: I wonder what type of structure we can see attached to a larva – is it a parazitoid larva?

I think it would be wonderfull to introduce the genus Asphondylia by placing it into the phylogenetic context of the family Cecidomyiidae, see Dorchin et al 2019 or Sikora et al 2019 (Dorchin N., Harris K.M. & Stireman III J.O., 2019: Phylogeny of the gall midges (Diptera, Cecidomyiidae, Cecidomyiinae): Systematics, evolution of feeding modes and diversification rates. Molecular phylogenetics and evolution 140: 106602; Sikora T., Jaschhof M., Mantič M., Kaspřák D. & ševčík J., 2019: Considerable congruence, enlightening conflict: molecular analysis largely supports morphology-based hypotheses on Cecidomyiidae (Diptera) phylogeny. Zoological Journal of the Linnean Society 185(1): 98–110).

Reviewer 2 Report

This is a good manuscript on gall midges of the genus Asphondylia associated with flower buds of Lamiaceae in Italy and reports two new species to science. 

A major problem in this manuscript is the similarity of COI sequences in A. nepeta and A. micromeriae. Firstly, the pairwise distance between each species and in each sequenced DNA region should be reported. Secondly, it seems that COI sequences of A. nepeta and A. micromeriae are similar because of contamination during the molecular experiments. There is no such case of similarity in the COI region in species that have clear morphological differences. COI is a good tool to separate Asphondylia species and this is very clear in the case of the closely related Japanese Asphondylia species which lack morphological differences, but differ in their life history and hosts. In addition, there are many studies on gall midges that included the usage of COI sequences, and non of them reported such an odd case in different species. I recommend the authors consider re-performing the molecular analysis to correct this result. 

The description parts of this manuscript are generally good but include comparisons to different species. I recommend the authors keep the descriptions sharp and transfer any comparisons to the comments section. I also inserted several minor suggestions in the attached revised manuscript.

The figure plates are beautiful, but I recommend you add extra figures to show the frontal view of the pupal head. This can help us know the relative length of the pupal horns and how they are arranged. 

Reviewer 3 Report

Section 2.1:  Sampling procedures are vague and nonspecific. More specific details are needed for other workers to replicate and interpret the work. For example:

How were samples selected (randomly? haphazardly?).

How were samples collected (by cutting the stem?).

How much of the stem was collected?

Please describe “monthly sampling” (i.e., when in the month were samples collected—first of the month, or middle of the month, or another time?).

Were samples collected in all years of the study?  

Please quantify “Some specimens” (line 84) and “more intensive sampling” (lines 82-83). What kind of “bags or boxes” (plastic? paper?...)?

Please explicitly state what is meant by “room temperature” and “normal photoperiod” (line 90).

Please state whether dissections were performed with or without a microscope.

Authors need to describe how the adults were “chosen for molecular analysis” and also provide sample sizes (lines 92-93).

“Several” (line 94) should be quantified.

Section 2.3:  Sample sizes need to be given in all cases.

Results:  Please provide the number of successful DNA amplifications/sequencings.  Were all adults selected for molecular analysis successfully amplified and sequenced?

Species descriptions:  “Antenna 2+12 segmented” (lines 184-185, line 195, line 278, and line 290):  Insects have only 3 antennal segments (scape, pedicel, and flagellum). The flagellum does not consist of segments. It consists of flagellomeres. Please make the correction (e.g., Antenna with 12 flagellomeres). 

Line 213: “As in other species” is not adequate for description of a new species. Please provide an explicit description.

Figure 4a:  The last-instar larva appears to have another, very small larva on it. Please explain.

Table 2: All percentages should be carried to the same decimal place.  If the authors are reporting 6.6, 3.3, etc., they should report 35.0, 8.0, 16.0, etc. (not 35, 3, 16…)

Line 354:  “diacritical characters”  This term does not make sense.

Line 395: “This hypothesis was verified”  Please state the hypothesis.  It is not clear what the authors are referring to as the hypothesis. (The previous sentence has 2 ideas [“hypotheses”], one about saprophytic growth and another about progressing in time).

Line 402:  “It is interesting to note”  This expression should be deleted.  It is subjective (and everything is interesting).

Lines 412-414:  Authors suggest that A. hornigi and A. serpylli are synonyms. However, no further statements are made and the readers (and taxonomists) are not given any guidance as to whether the two names are actually being synonymized or not.  Please provide more explanation about this suggestion.

Lines 463-466”  The sentence beginning with “The occasionally simultaneous isolation” is poorly written and confusing.  Please rewrite.  The sentence that follows this sentence is also poorly written.

The paper contains numerous typographical errors and examples of awkward or incorrect English language. Authors should carefully read their paper and have a native English speaker assist.

Reviewer 4 Report

The work of Bernardo and collaborators concerns a study conducted on diptera of the cecidomyidae family with the description of two new species for science together with notes on the bioecology of the aforementioned dipterans. The work was carried out with remarkable scientific rigor and well set up in the various chapters. I found the tables shown in the text well composed and very interesting, which clearly indicate the differences at a specific level. I believe that the work can be published after few minor changes which I noticed in a separate file 
